# Radiotherapy—Dose Escalated for Large Volume Primary Tumors—And Cetuximab with or without Induction Chemotherapy for HPV Associated Squamous Cell Carcinoma of the Head and Neck—A Randomized Phase II Trial

**DOI:** 10.3390/cancers15092543

**Published:** 2023-04-28

**Authors:** Claes Mercke, Gun Wickart-Johansson, Helena Sjödin, Pedro Farrajota Neves da Silva, Gabriella Alexandersson von Döbeln, Gregori Margolin, Sara Jonmarker Jaraj, Hanna Carstens, Anders Berglund, Ingmar Lax, Mats Hellström, Lalle Hammarstedt-Nordenvall, Signe Friesland

**Affiliations:** 1Theme Cancer, Medical Unit Head&Neck, Lung and Skin Cancer, Karolinska University Hospital, Comprehensive Cancer Center, 17176 Stockholm, Sweden; 2Department of Oncology-Pathology, Karolinska Institutet, 17176 Stockholm, Sweden; 3Clinical Pathology and Cancer Diagnostics, Karolinska University Hospital and Department of Oncology-Pathology, Karolinska Institutet, 17176 Stockholm, Sweden; 4Department of Clinical Sciences, Intervention and Technology, Karolinska Institutet, 17164 Stockholm, Sweden; 5Division of Ear, Nose and throat Diseases and Department of Clinical Sciences, Intervention and Technology, Karolinska Institutet, 17176 Stockholm, Sweden; 6Department of Neuroradiology, Karolinska University Hospital, 17176 Stockholm, Sweden; 7Epistat Epidemiology and Statistics, 75440 Uppsala, Sweden; 8Theme Cancer, Karolinska University Hospital, 17176 Stockholm, Sweden

**Keywords:** head and neck cancer, radiotherapy, radiobiology, radiosensitizer, distant metastases

## Abstract

**Simple Summary:**

Most patients with HPV associated squamous cell carcinoma of the head and neck, treated with chemoradiotherapy, CRT, for cure, and who succumb to their disease, do so because of distant metastases. Even if a local recurrence as first site of treatment failure is rather low in this disease, characterised as being radiosensitive, such failures are still a problem. In this randomized phase II study we aimed to exploit whether induction chemotherapy IC, with two cycles of Taxotere, Cisplatin and 5-FU, TPF, could improve progression -free survival, PFS, by diminishing the rate of distant failures, administered before CRT. To reduce risk of toxicity from cumulated doses of cisplatin, concurrent chemotherapy during RT was given with cetuximab. To improve local control and mitigate the negative impact of a large tumor volume, an escalated RT dose was prescribed for patients with such tumors. PFS was found to be similar in the two study arms. However, even if not statistically significant, there were twice as many patients with distant relapses in the group of patients who had not IC. Overall survival was high and local recurrence as first site of failure was low, in 4.6% of patients and was similar for T1/T2 and locally bulky T3/T4 tumors. An escalated RT dose might have mitigated the negative impact of a large tumor volume but for some patients even this intensified treatment was insufficient. With respect to patients’ response to IC, 29% of the patients in this treatment arm could be identified to have no relapse whether locoregional or distant during the time of follow up. These data shed further light on the characteristics of this disease of importance for the planning of future studies. If induction chemotherapy with cisplatin could reliably help to select patients for de-escalated subsequent RT, cetuximab is considered a good candidate to be given concurrently with RT to diminish morbidity from cumulated doses of cisplatin. The role of escalated RT doses for selected patients with bulky T3 and T4 tumors, perhaps primarily for those who do not respond to IC should also be addressed in future studies.

**Abstract:**

The leading cause of death for patients with HPV associated squamous cell carcinoma of the head and neck (SCCHN) after treatment with chemoradiotherapy (CRT) nowadays is peripheral metastasis. This study investigated whether induction chemotherapy (IC) could improve progression free survival (PFS) and impact on relapse pattern after CRT. Methods: Eligible patients in this multicenter, randomized, controlled, phase 2 trial had p16-positive locoregionally advanced SCCHN. Patients were randomized in a 1:1 ratio to either RT with cetuximab (arm B) versus the same regimen preceded by two cycles of taxotere/cisplatin/5-FU (arm A). The RT dose was escalated to 74.8 Gy for large volume primary tumors. Eligibility criteria included patients of 18–75 years, an ECOG performance status 0–1, and adequate organ functions. Results: From January 2011 to February 2016, 152 patients, all with oropharyngeal tumors were enrolled, 77 in arm A and 75 in arm B. Two patients, one in each group, withdrew their consent after randomization, leaving 150 patients for the ITT analysis. PFS at 2 years was 84.2% (95% CI 76.4–92.8) in arm A and 78.4% (95% CI 69.5–88.3) in arm B (HR 1.39, 95% CI 0.69–2.79, *p* = 0.40). At the time of analysis, there were 26 disease failures, 9 in arm A and 17 in arm B. In arm A, 3 patients had local, 2 regional, and 4 distant relapses as first sites of recurrence, and in arm B, 4, 4, and 9 relapses in corresponding sites. Eight out of 26 patients with disease progression had salvage therapy and 7 were alive NED (no evidence of disease), at 2 years. Locoregional control was 96% in arm A and 97.3% in arm B and OS 93% and 90.5%, respectively. Local failure as first site of recurrence was low, in 4.6% of patients and was similar for T1/T2 and T3/T4 tumors (n.s). Nevertheless, out of 7 patients with primary local failures, 4 were treated with the escalated RT dose. Toxicity was low and similar in the treatment arms. There was one fatal event in arm A where the combined effects of the drugs used in chemotherapy and cetuximab could not be ruled out. Conclusions: PFS, locoregional control and toxicity did not differ between the two arms, OS was high, and there were few local relapses. In arm B, more than twice as many patients had distant metastasis as the first site of relapse compared to arm A. The response to IC was found to define 29% of patients in arm A who did not have a tumor relapse during follow-up. An escalated dose of 74.8 Gy could mitigate the negative impact of large tumor volume but for some patients, even this intensified treatment was insufficient.

## 1. Introduction

For patients with locoregionally advanced squamous cell carcinoma of the head and neck, radiotherapy, RT, with concurrent cisplatin is a standard treatment. Most patients who fail such chemoradiotherapy (CRT) do so because of a local or regional relapse. Therefore, most trials studying methods to improve outcomes for this group of patients emphasize locoregional disease control.

HPV positivity has emerged as the most influential determinant of survival for patients with their tumors located in the oropharynx and patients with an HPV-driven oropharyngeal cancer have a risk of death of about half of their HPV unrelated counterparts. An 8-year overall survival rate, (OS), of >70% has recently been reported for patients with HPV positive tumors versus about 30% for patients with HPV negative tumors [1,2]. The favorable prognosis has led the medical community to implement a new TNM staging system with respect to HPV association but has also raised interest in finding ways to optimize treatment for these patients and to avoid unnecessary long-term morbidity.

Most studies on HPV positive cell lines from SCCHN indicate a higher intrinsic cellular radiosensitivity compared to HPV-negative strains, mainly accredited to diminished repair of DNA double strand breaks [3,4]. Thus, the improved outcome for patients with HPV positive tumors has been attributed to increased locoregional control of the disease. With the rarity of locoregional failure, distant metastasis is now the leading cause of death in HPV-positive patients [5,6]. To reduce the risk of long-term treatment-induced toxicities, several trials are examining whether regimens with de-escalated intensity can be given without compromising the favorable outcome with respect to tumor control. These studies have mainly focused on radiotherapy dose or volume reduction, modification of concomitant chemotherapy, the use of surgery as a definitive treatment, or finally the response to induction chemotherapy to de-escalate locoregional (chemo)radiotherapy or surgical resection [7].

At the time of trial preparation, retrospective studies indicated that chemotherapy given concomitantly with RT did not diminish the incidence of distant metastases in this group of patients, which is supported by recent findings [8].

The present study was designed as a prospective, randomized phase II trial to compare outcome and recurrence patterns, with specific reference to the site of first recurrence, between patients who were treated with RT with concomitant cetuximab and patients who received the same regimen preceded by TPF induction chemotherapy (IC), a combination of Taxotere, cisplatin, and 5-FU.

Cetuximab was chosen to be given concurrently with RT to avoid undue toxicity from cumulative doses of cisplatin in the group of patients randomized to IC. To minimize the risk of locoregional failure as the first site of recurrence, the RT dose was escalated for large volume tumors of stage T3-T4 and T2 tumors of the base of the tongue, with the largest diameter being 3–4 cm.

## 2. Materials and Methods

### 2.1. Study Design

The ACCROBAT study (“accelerated, concurrent chemoradiotherapy with brachytherapy”), was an open-label, double-blind, randomized controlled phase 2 study conducted in Sweden with 3 participating centers, Karolinska University Hospital and Södersjukhuset in Stockholm and Uppsala University Hospital. Patients were randomly assigned 1:1 between the treatment groups, and the randomization was stratified with respect to non-smokers vs. smokers/former smokers. Randomization codes were generated centrally at the Clinical Trial Unit of the Karolinska University Hospital.

### 2.2. Objectives

The primary objective of the study was to investigate progression-free survival (PFS) in patients treated with RT plus cetuximab (study arm B) compared with the same regimen preceded by 2 cycles of IC (study arm A). Secondary objectives were to compare the recurrence pattern with respect to the first site of recurrence, locoregional control, overall survival (OS), and toxicity, between the treatment groups (Figure 1).

### 2.3. Patients

Eligible patients were 18–75 years, had histologically confirmed, previously untreated, non-resectable squamous cell carcinoma of the oral cavity, oropharynx, hypopharynx, or larynx, stage III to IV, excluding stages T1N1 and T2N1, according to UICC TNM classification, 7th edition. Resectability was defined as conditions making it possible to extirpate the tumor without undue mutilation (Table 1). Even if oropharyngeal tumors could fulfill such conditions, RT is considered the treatment of choice for these tumors with respect to functional outcome and quality of life in our institution. Patients had no distant metastases, had an HPV positive tumor, and a WHO/ECOG performance score of 0–1. HPV positivity was defined as tumors positive for p16 on immunohistochemistry and, when doubtful, completed with PCR. Involved laboratories at each participating hospital were accredited and certified. Patients had to fulfil the following inclusion criteria: measurable disease according to RECIST version 1.1 [9], a life expectancy of at least 3 months, and recorded smoking history. Participants were excluded if they had previous malignancies, had active, serious underlying medical conditions such as ongoing infections, a pre-existing history of severe lung disease, or severe or uncontrolled cardiovascular disease, including myocardial infarction within the last twelve months. Laboratory tests to verify adequate renal, bone marrow, and liver function were required. A CT scan or MRI, or both, of the head and neck region, and chest CT and an optional 18F-fluorodeoxyglucose PET scan were also performed. All patients received oral and written information about the study and signed a written consent before study inclusion. The study was approved by the regional ethics committee in Stockholm and registered at ClinicalTrials.gov (EudraCT:2009-013438-26).

### 2.4. Treatment

#### 2.4.1. Chemotherapy

In arm A, patients received two cycles of IC with 75 mg/m^2^ docetaxel plus cisplatin 75 mg/m^2^, both drugs infused over 1 h together with 1000 mg/m^2^ of 5-FU administered in a continuous infusion over 24 h, given 21 days apart. For patients with severe hearing loss, cisplatin could be replaced with a 30 min infusion of carboplatin (area under the curve = 6).

Dose modifications for IC were permitted if patients developed neutropenia or thrombocytopenia. If patients experienced febrile neutropenia, or absolute neutrophil counts <0.5 × 109/L for more than one week, their next treatment cycle was postponed until recovery and the next course reduced by 25%. Dose reductions were also allowed if patients developed high-grade hepatic or neuropathic adverse events or myalgia. Cetuximab was infused at a dose of 250 mg/m^2^ over 60 min weekly during RT to all patients after prior premedication with antihistamine and a steroid.

A start-up dose of 400 mg/m^2^ of cetuximab, infused over 2 h, was delivered to all patients before the start of RT.

#### 2.4.2. Radiotherapy

In arm A, RT target volumes were defined by the extent of disease before chemotherapy and adjusted to conform to anatomy after chemotherapy. Conventional definitions of targets for the prescriptions of different dose levels were chosen according to Table 2. The prescribed dose for the tumor volumes was chosen as the dose of which 95% was found to best fit the isodose to the PTV, with minor exceptions when dose to organs at risk (OAR) had higher priority. All patients were planned with intensity-modulated RT or volumetric modulated arctherapy and treated with 6 MV photons. RT could be delivered with different schedules, either as sequential or as simultaneous integrated boost (SIB), as shown in Table 2. The brachytherapy boost (BT) was generally delivered with pulsed dose rate RT (PDR). Schedules 1 and 2 as well as 3, 4, 5, as seen in Table 2, were prepared to give similar biological effects, as determined with the LQ formalism. Schedules 3, 4, and 5 were used for T3, T4, and bulky tumors of the base of tongue. The CTV in the T-position, where dose was escalated, was the GTV, delineated as a separate target on the treatment planning CT scan. The external beam RT was delivered with an accelerated schedule with 6 fractions per week. On one weekday, 2 fractions were delivered at least 6 h apart. Dose volume constraints for organs and tissues at risk were specified according to local guidelines.

### 2.5. Procedures during Study/Follow-Up

Tumor response was assessed with CT or MRI after the completion of IC in arm A, 6–8 weeks after the completion of CRT and after 1 and 2 years in all patients. PET scan imaging with CT was performed if the interpretation of CT or MRI was inconclusive, or to identify new lesions. The response was graded according to RECIST version 1.1 guidelines [10]. The physical examination, including fibre-optic endoscopy, was done at baseline before treatment, after IC, 6–8 weeks after completed CRT and at follow up every 3 months for the first 2 years and every 6 months for year 3–5. Assessments of symptoms and adverse events, AE:s, were monitored continuously and graded according to CTCAE v4.0 [11]. The cut-off for early versus late AE:s was set at 90 days after the completion of CRT. Serious AE:s (SAE) were reviewed throughout the study by the principal investigator.

All patients with residual disease or disease progression were evaluated for salvage therapy, surgery, reirradiation, or pharmacotherapy.

### 2.6. Statistical Analysis

The primary endpoint was differences in PFS at 2 years, defined as the time from the first day of treatment until date of disease progression or death from any causes. If patients died without disease progression, survival data were censored at the last visit alive. Secondary endpoints were OS, defined as the time from treatment initiation to death from any cause, locoregional tumor control, i.e., absence of tumor progression at irradiated sites, response rate, pattern of relapse, and distant metastasis-free survival. The intention to treat (ITT) population was used for efficacy analyses and contains all patients meeting eligibility criteria. The PFS curves in both treatment arms were compared using a non-inferiority log rank test. Kaplan–Meier methods were used to estimate the PFS curves. Univariate Cox proportional hazard methods were used to estimate the hazard ratio and corresponding 95% CI between the two treatment arms.

### 2.7. Sample Size Calculation

Based on current clinical experiences at the time of start of the study, a PFS rate of 70% was expected in HPV positive patients in the control arm at 2 years. The study was not powered to show non-inferiority of the study arm in comparison to the control arm, using a sufficiently small non-inferiority margin and a one-sided type-I error of 0.025. However, when the sample size in each randomized arm is 75, with a total number of events required (E, progression or death) = 70, a log-rank test of non-inferiority of the survival curve for the experimental arm B to the survival curve for the control arm A with a 0.2 one-sided significance level will have 80% power to reject the null hypothesis of inferiority (a hazard ratio of 1.5 or greater) when the arm A exponential hazard rate is 0.1783/year (equivalent to a PFS rate at 2 years of 70%) and the true hazard ratio is 1.0, assuming an accrual period of 2 years, a maximum follow-up time of 5 years, and a dropout of 5% per year in each arm.

## 3. Results

### 3.1. Treatment Outcome

From 24 January 2011 to 23 February 2016, 152 patients, all with oropharyngeal cancer, were enrolled across three sites, 77 patients in arm A and 75 in arm B. Two patients, one in each group, withdrew their consent after randomization, leaving 150 patients for the ITT analysis, 76 in arm A and 74 patients in arm B. The cutoff was 15 July 2018. Patient characteristics were well balanced between groups (Table 1).

At the time of analysis, median follow-up was 39.3 months (interquartile range (IQR) 37.7–41.5 months) in arm A and 39.3 months (IQR, 37.9–41.3 months) in arm B. There were 26 disease failures among all patients, 9 in arm A and 17 in arm B. In arm A, there were 3 local, 2 regional, and 4 distant relapses as the first site of recurrence and in arm B, 4, 4, and 9 relapses in corresponding sites. One patient in arm B had a simultaneous distant and local relapse. Local recurrence as the first site of failure was therefore seen in 7/152 patients (4.6%), was not related to the T stage, was seen in 0/34 T1 tumors, 4/68 T2 tumors, 1/27 T3 tumors, and in 2/21 T4 tumors (n.s.).

Twice as many patients in arm B (9/74 = 12%) compared to arm A (4/76 = 5.2%) had distant metastasis as the first site of recurrence (n.s.). The site of the original tumor, tonsil or base of tongue, and their TNM stages in relation to the site of first relapse are shown in Table 3.

PFS at 2 years was 84.2% (95% CI 76.4–92.8) in arm A and 78.4% (95% CI 69.5–88.3) in arm B (HR 1.39, 95% CI 0.69–2.79, *p* = 0.40), Figure 2. There were no differences regarding sites of first recurrence, local, regional, or distant, between the two study groups, as seen in Figure 3 (*p* = 0.1 and 0.5 respectively).

At the time of analysis, 19 patients had died, 9 in group A and 10 in group B, and all patients had disease progression as their cause of death. When a relapse had been diagnosed, patients were offered salvage therapy, surgery, radiotherapy, or pharmacotherapy. Eight out of the 26 patients with relapses were treated with an intention to cure and 7 were alive with NED at the time of follow up at 2 years. This was the result of the salvage treatment given to one patient with a local failure, five patients with regional failure and one patient with a metastatic lung lesion put on experimental chemotherapy. Local control at 2 years was 95.9% in arm A, locoregional control 96%, and distant control 95.9% in arm A and 95.7, 97.3, and 91.9% in arm B. OS at 2 years was 93.4% for patients in arm A and 90.5% in arm B.

In arm A 73/76 patients were evaluable for response to IC according to the protocol. On T site, 28/73 (38%) patients had a complete tumor response, CR, and 32/73 (44%) a partial response, PR. Stable disease, SD, was registered in 11/73 (15%) and progressive disease, PD, in 2/73 (3%). On the N site, CR was 29/73 (40%), PR 26/73 (36%), SD 15/73 (20%), and PD 3/73 (4%).

CR on both sites was observed in 14/73 patients, and CR + PR in 8/73 patients. Of those, only one of the patients who had a response on the T and N site (CR + CR or CR + PR) had recurring disease. This response pattern to IC could therefore identify 21 out of 73 patients (29%) who never had a recurrence during follow up.

### 3.2. Toxicity

There was one fatal event in the study. This patient was allocated to treatment arm A and was suddenly hospitalized because of hypoxia (grade 5). No certain cause of death was identified but a combined effect of the drugs in IC and cetuximab could not be ruled out. All other AE:s in grade 3–4, were anticipated and transient. They could be taken care of and managed adequately according to protocol and local guidelines. Additionally, most side-effects were related to RT. The toxicity profile of patients experiencing AE:s was similar in both arms with one important difference: as expected, febrile neutropenia was only seen in arm A, and three patients in this arm had to be treated with antibiotics. Two cycles of TPF were given according to the protocol to 64/76 (84%) of patients and one cycle to the rest of patients because of the reduced neutrophil count. Diarrhea was seen in seven patients in arm A, fatigue in eight patients, and six in arm A, which was considered as side effect of IC. Transient nausea was also registered in 14 patients, whereof 10 were in arm A, and was also considered as an effect of IC.

The most frequent AE was mucositis of the oral cavity or the oropharynx, reported in 50 and 46 patients, respectively, in the two arms. In no patient did this acute toxicity increase treatment time. However, some individuals had more serious toxicity than required shorter hospital stays. An acneiform rash was seen in low frequency, in 17 patients in arm A and in 15 in arm B, judged as an AE due to cetuximab. The rash was described as annoying and sometimes painful but subsided spontaneously in all patients. Hypersensitivity allergic reactions to cetuximab were rare and seen in 3% of patients. No patient with grade 3–4 acute toxicity was registered regarding hearing, tinnitus, acute kidney injury, or neuropathy. Late toxicity was registered with the RTOG scale for late side-effects after RT [12]. The frequency of grade 3 xerostomia and dysphagia was low and in line with previous observations, with a similar distribution between treatment groups. Trismus was not registered in any patient, but osteoradionecrosis was seen in one patient in arm B.

## 4. Discussion

To our knowledge, this is the first prospective, randomized trial to study the effect of IC on PFS and recurrence pattern in patients with locally advanced HPV-associated SCCHN treated with definitive CRT, where the dose of RT was escalated for bulky primary tumors. PFS, which was the primary parameter of the study, did not differ between the two study groups and was in line with recent reports. Patients with SCCHN historically developed locoregional recurrence as the primary site of treatment failure. Even after adjusting for age, tobacco use, and performance status, HPV association in patients with oropharyngeal tumors confers a survival advantage compared to HPV-negative disease. HPV-positive head and neck tumors are characteristically more radiosensitive compared to their HPV-negative counterparts, and the insufficient repair of radiation-induced DNA lesions conferred by HPV-positivity could be one basic mechanism [10]. In the present study, local relapses were remarkably few, in 7/152 patients (4.6%), in line with other reports, and with no differences between the two treatment arms. Moreover, the number of local relapses in T3 and T4 tumors, generally considered as “intermediate or high-risk tumors”, were not more frequent than in T1 and T2 tumors. This is in line with recent findings, where the outcome of patients with oropharyngeal cancer treated with RT alone was largely determined by tumor volume, even when adjusting for other established prognostic factors and even more pronounced in patients with p16-positive tumors [13]. The hypothesis that intensified RT could mitigate the negative prognostic impact of large tumor volume in patients with HPV-positive tumors is therefore supported by the findings in the present study. Importantly, positive results from more intensive RT were achieved without an increase in toxicity. The fact that four out of seven patients with local treatment failure were treated with the escalated RT dose, means that there are patients with HPV positive tumors who need even more intensified treatment regimens to be locally controlled.

Distant metastasis is now considered the leading cause of death for these patients, so it is reasonable to hypothesize that patients could gain from regimens including systemic therapy. RT given simultaneously with cisplatin, CRT, is one standard of care for patients with locally advanced SCCHN, both as definitive treatment and after surgery when pathological adverse features are identified. The addition of IC to CRT for these patients has been studied in several randomized clinical trials but the majority has not demonstrated a survival benefit [14,15,16]. These previous studies have included heterogenous patient populations with many head and neck subsites, with the application of a variety of IC regimens and with both HPV positive and negative tumors. As illustrated with results from the MACH-NC group, there was no survival benefit with IC added to CRT, but a reduced distant failure rate [17]. HPV positive oropharyngeal tumors represent a highly chemo-sensitive disease [18,19]. The potential effect of IC for such a disease entity is illustrated in a study where induction TPF, as used in the present study, was added before CRT for locally advanced nasopharyngeal carcinoma, which resulted in superior distant control, improved disease-free survival, and OS [20].

When this study was planned, there were no outcome data comparing RT delivered concurrently with either cisplatin or cetuximab for patients with HPV positive oropharyngeal cancer. The Bonner study evaluated the outcome of RT together with cetuximab versus RT alone. The investigators found outcome advantages with the combined regimen also in the subgroup of patients with HPV-positive tumors [21]. Recent randomized prospective phase III studies, where only patients with HPV-positive oropharyngeal cancer were included, reported inferior locoregional control with cetuximab compared to cisplatin given concurrently with RT [22,23]. Nevertheless, outcome data at 2 years in the present report are in line with those presented in these randomized studies. The reason for choosing cetuximab as a radiosensitizer was based upon data showing a reasonably mild toxicity in earlier studies, as in the Bonner study, and the lack of additional toxicity when given together with platinum-based chemotherapy for patients with recurrent tumors documented in the EXTREME regimen [24]. Since cisplatin was a major component of the IC in the present study, the possibility to use cetuximab as a radiosensitizer was important to avoid major toxicity from cisplatin. In the above-mentioned randomized phase III studies comparing concurrent cisplatin and cetuximab for HPV positive oropharyngeal cancer, the differences in toxicity profile were small but did not favor cetuximab [25]. Toxicity in the present study was mild with no patient experiencing renal toxicity or any kind of neuropathy. This could be of importance if a definitive role of cisplatin containing IC to counteract risk of distant metastasis is established for patients with HPV related oropharyngeal cancer. In the present study, even if not statistically significant, there were twice as many patients with distant relapses as first site of recurrence in the group of patients who had not IC and time to distant failure was shorter in this treatment arm (Figure 3). Moreover, more than 50% of all failures were at distant sites. Considering earlier data, a follow up of 2 years, as in the present study, may be too short to identify the full potential of these tumors inherent biology to metastasize distantly. The role of cisplatin-based IC to diminish risk of distant failure should therefore be examined with a longer follow up of the present patient material.

The response of the individual patients’ tumor to IC has been used as a marker of radiation sensitivity to select patients for a reduced RT dose [26,27]. Beyond radiation dose de-escalation the role for radiation volume de-escalation was studied in the so-called OPTIMA trial where acute toxicities were significantly reduced and where tumor control was excellent [28]. To de-escalate treatment intensity without jeopardizing excellent treatment outcomes is much-desired for these reasonably young patients who may live long with treatment induced sequelae. However, data from reported various phase II trials are promising but evidence from phase III studies are either still lacking or has failed to demonstrate comparable outcomes for de-escalated regimens. In the present series there were 21/73 (29%) patients who showed a well-defined response pattern to IC and who did not have a tumor recurrence during the follow up of 2 years. It is a reasonable hypothesis that such patients might be candidates for de-escalated therapy regimens with reduced intensity of RT to some volumes of tumor [28]. If cisplatin-containing IC could diminish both risk of distant metastasis and select patients for an amelioration of RT, it is important to know, as shown in the present study, that cetuximab delivered together with subsequent RT does not affect the risk of serious complications sometimes seen with cumulated higher doses of cisplatin. However, the role of an escalated RT dose for selected patients with bulky T3 and T4 tumors, perhaps primarily for those who do not respond to IC, should also be addressed in further studies. The fact that most patients with HPV associated squamous cell carcinomas who succumb from their disease do so because of distant metastasis supported by data in the present study. Therefore, it is important to find measures to improve the effects of systemic therapy for this disease. The role of immunotherapy is a new paradigm in this setting. Immunotherapy has been incorporated into programs relying on definitive chemoradiation, as in induction chemoimmunotherapy regimens as well as in adjuvant treatments. Preliminary results have demonstrated excellent oncologic outcomes with reduced toxicity [7]. In a recent study, the effect of induction therapy with a single dose of double immune checkpoint blockade immediately after a single-cycle platinum and docetaxel as was compared to chemotherapy alone regarding complete remission rates. The investigators found a complete remission rate of 60.3% after chemoimmunotherapy compared with 40.3% after chemotherapy alone [29]. We think that the main limitation of the present study is the short follow up of the patients, a time which is too short to show in a more correct way the tendency of the tumors to recur, both locoregionally and at distant sites. However, short-term data as presented in this study should have an impact on hypotheses related to the planning of future studies and a long-term follow up of the present study is also under way.

## 5. Conclusions

RT with concurrent cetuximab, with or without IC, gives PFS, locoregional tumor control, and overall survival in line with the best results in the literature for patients with HPV associated oropharyngeal tumors. IC did not change PFS or the relapse pattern of the disease. However, at 2 years of follow up, there were more than twice as many distant relapses in the group of patients who did not have IC. Moreover, more than 50% of all patients had their first site of relapse at distant sites, underscoring the importance of finding effective systemic treatments. If IC with cisplatin could reliably help to select patients for de-escalated subsequent RT, cetuximab is considered a good candidate to be given concurrently with RT to diminish risk of morbidity from cumulated doses of cisplatin. The finding that local relapses were not more frequent in patients with large volume primary tumors supports the value of studying the role of escalating the RT dose to such tumors in the future.

## Figures and Tables

**Figure 1 cancers-15-02543-f001:**
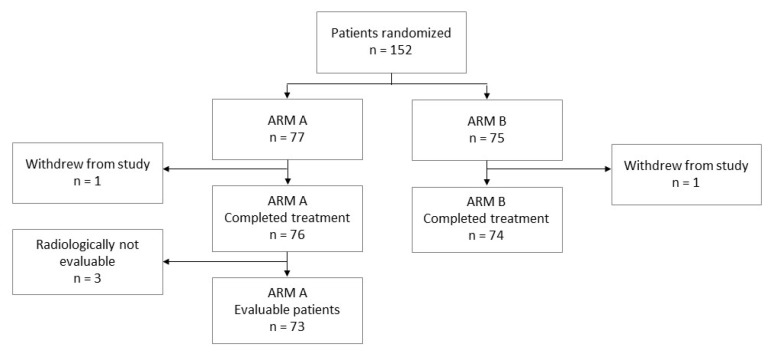
Study flow chart. ARM A: induction chemotherapy (IC) followed by radiotherapy (RT) plus cetuximab; ARM B: radiotherapy (RT) plus cetuximab.

**Figure 2 cancers-15-02543-f002:**
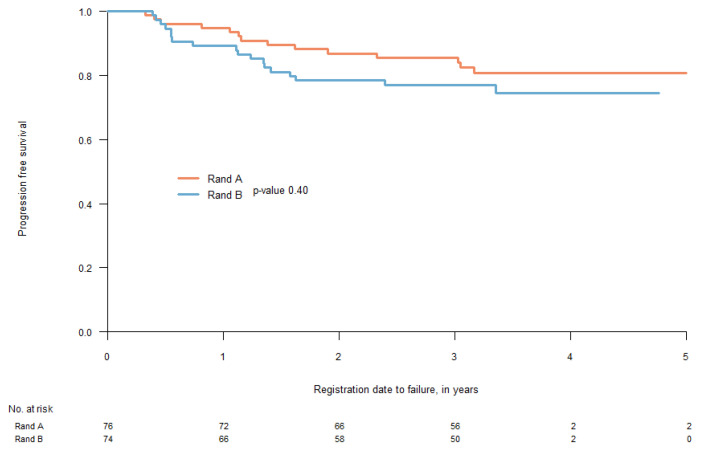
Progression free survival at 2-years was 84.2% (95% CI 76.4–92.8) in arm A and 78.4% (95% CI 69.5–88.3) in arm B (HR 1.39, 95% CI 0.69–2.79, *p* = 0.40).

**Figure 3 cancers-15-02543-f003:**
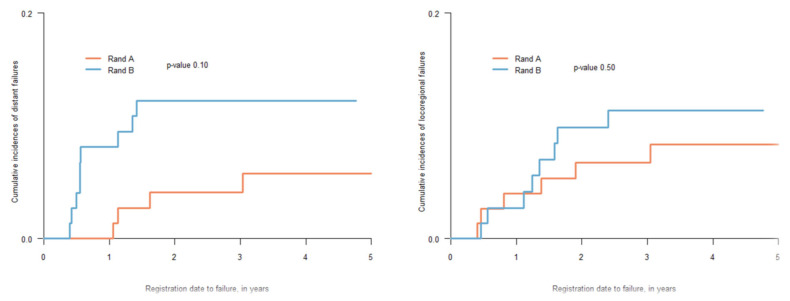
Cumulative incidence of distant failure (HR 2.40, 95% CI 0.74–7.79), locoregional failure (HR 1.40, 95% CI 0.49–4.04).

**Table 1 cancers-15-02543-t001:** Baseline Demographics, clinical characteristics, by randomized group for the ITT population (*n* = 150).

		Total	Arm A	Arm B
Randomised		150	76	74
Age:		40.3–80.7 Median 59.8	40.3–80.7 Median 59.8	40.7–78.7 Median 59.7
Sex:	Female	37 (24.6%)	20	17
	Male	113 (75.4%)	56	57
Smoking	Never smoker Smoker/former smoker	35.3% 64.7%	22 54	31 43
PS:	0	143 (95.3%)	72	71
	1	7 (4.7%)	4	3
Tumor site:	Oropharynx, tonsil	95 (63.3%)	51	44
	Oropharynx, base of tongue	55 (36.7%)	25	30
T-stage:	T1	34 (22.7%)	16	18
	T2	68 (45.3%)	37	31
	T3	27 (18%)	15	12
	T4	21 (14%)	8	13
N-stage:	N0	8 (5.3%)	3	5
	N1	8 (5.3%)	4	4
	N2	134 (89.3%)	69	65
T + N-stage:	T1N0	0	0	0
	T1N1	0	0	0
	T1N2	36	17	19
	T2N0	2	0	2
	T2N1	1	0	1
	T2N2	65	37	28
	T3N0	3	2	1
	T3N1	4	3	1
	T3N2	18	9	9
	T4N0	3	1	2
	T4N1	3	1	2
	T4N2	15	6	9

**Table 2 cancers-15-02543-t002:** Doses given i Gy. Number of fractions within parantheses.

Schedule	GTV	CTV1, Lymphnode Metastases	CTV2, Elective Volumes	Delivery
1	68 (34)	68 (34)	46 (23)	Sequential
2	68 (34)	68 (34)	51.7 or 54.4 (34)	SIB
3	74.8 (34)	68 (34)	51.7 or 54.4 (34)	SIB
4	68 (34) + 8-10 BT	68 (34)	51.7 or 54.4 (34)	SIB
5	68 (34) + 8-10 BT	68 (34)	46 (23)	Sequential

GTV = gross tumor volume, CTV1 = clinical target volume containing gross tumor plus safety margin, CTV2 = clinical target volume containing microscopic tumor plus safety margin, BT = brachytherapy, SIB = simultaneous integrated boost technique.

**Table 3 cancers-15-02543-t003:** Site of primary tumor, TNM stages I relation to site of first relapse.

	Local	Regional	Distant
ARM A	[*n* = 3]	[*n* = 2]	[*n* = 4]
	T3N2b Base tongue	T2N2b Tonsil	T2N2b Tonsil
	T2N2c Tonsil	T2N2b Tonsil	T4N2c Base tongue
	T4N2b Tonsil		T2N2b Tonsil
			T2N2c Tonsil
ARM B	[*n* = 4]	[*n* = 4]	[*n* = 9]
	T2N2b Tonsil	T2N2b Tonsil	T3N2b Base tongue
	T2N2b Tonsil	T2N2b Tonsil	T4N2c Tonsil
	T4N2b Base tongue	T2N2b Base tongue	T4N2c Tonsil
	T2N2b Base tongue	T1N2b Base tongue	T2N2b Tonsil
			T3N2b Tonsil
			T2N2b Base tongue
			T1N2b Tonsil
			T4 N2b Base tongue
			T4N0 Tonsil

## Data Availability

The data presented in this study are available on request from the corresonding authors, but cannot be made publicly available due to Swedish laws and personel confidential informations.

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
