# Peer review of "Radiotherapy—Dose Escalated for Large Volume Primary Tumors—And Cetuximab with or without Induction Chemotherapy for HPV Associated Squamous Cell Carcinoma of the Head and Neck—A Randomized Phase II Trial"

_cancers, 2023, doi:10.3390/cancers15092543_

Round 1

Reviewer 1 Report

This text describes a phase 2 clinical trial that aimed to compare two treatment regimens for patients with p16-positive locoregionally advanced squamous cell carcinoma of the head and neck (SCCHN). Eligible patients were randomized in a 1:1 ratio to either radiotherapy with cetuximab (arm B) or the same regimen preceded by 2 cycles of taxotere/cisplatin/5-FU (arm A). The trial found that there was no significant difference in progression-free survival (PFS) between the two arms. Locoregional control was high in both arms, and overall survival (OS) was similar as well. Toxicity was low and similar in both arms. Local failure as the first site of recurrence was low and similar for T1/T2 and T3/T4 tumors. While I may disagree with the assessment that there is utility in the escalated radiation dose given the distant failure rate, and the manuscript does not do a great job explaining how to improve upon the study with biomarkers, the results are otherwise well written and acceptable for publication.

Reviewer 2 Report

In the article "Radiotherapy – dose escalated for large volume primary tumors - and Cetuximab with or without induction chemotherapy for HPV associated squamous cell carcinoma of the head and neck – a randomized phase II trial." Mercke et al. presents the results of a phase II randomized trial that evaluated a treatment escalation strategy using 2 cycles of TPF induction chemotherapy (using Docetaxel and Cisplatin as taxanes and platinum salt respectively) followed by radio-biotherapy with Cetuximab in head and neck cancers p16-positive. It should be mentioned that, although there is the concept that these tumors are very radiosensitive compared to cases of head and neck cancer related to smoking, the authors anticipate a possible variation in radiosensitivity for large volume tumors and choose to escalate the dose up to 74.8Gy. The article also includes a flow chart that describes the study algorithm, 3 tables that present the characteristics of the patients, the fractionation regime of radiotherapy and the characteristics of the patients in relation to the first relapse. Also in 3 figures, the authors present PFS at 2 years in the two arms and cumulative incidence of distance failure and locoregional failure. Even if IC does not demonstrate the ability to influence PFS and pattern of failure, the authors mention the perspectives and open hypotheses of the study. The true value of the study is to highlight the "biomarker" value of IC in choosing a regimen of escalating the radiotherapy dose for bulky T3 and T4 or de-escalating the radiation dose and replacing Cisplatin with Cetuximab or reducing the radiotherapy dose in selected cases . The study demonstrates that p16 positive HNC are heterogeneous tumors both in terms of pattern of failure and sensitivity to radiotherapy/chemotherapy, so a refinement of therapeutic strategies for subgroups of p16 positive cases is necessary.

Reviewer 3 Report

Mercke et al. provide the results of a prospective randomized controlled multi-center phase II trial aimed at determining the difference in progression-free survival between induction chemotherapy (docetaxel, cisplatin, 5-FU) following by cetuximab + radiotherapy (Arm A) versus cetuximab + radiotherapy (without induction; Arm B) in patients with unresectable HPV-positive oropharyngeal cancer. The authors started the ACCROBAT study in 2007 (https://ascopubs.org/doi/abs/10.1200/jco.2014.32.15_suppl.e17004) and focus on HPV-positive disease in the current study, which was initiated in 2010 (https://ascopubs.org/doi/abs/10.1200/JCO.2019.37.15_suppl.6077).

It is curious as to why the present study is just being submitted for publication, as accrual finished in 2016, the 2-year primary endpoint would have been met in mid-2018, and 5-year follow-up finished in early 2021. The delay is unfortunate because, in the interim, the utility of induction chemotherapy in this setting has been brought into question (https://ascopubs.org/doi/10.1200/JCO.2016.68.3300?url_ver=Z39.88-2003) in addition to the use of cetuximab for chemoradiation (https://www.thelancet.com/journals/lancet/article/PIIS0140-6736(18)32752-1/fulltext). In any case, the data are what they are, and they may be a useful addition to the literature with some extensive revision and additional contextualization. There are many phase II trials in HPV+ HNSCC, and these need to be published in order to push the field forward in areas of uncertainty—and induction chemotherapy is certainly one of those areas.

Major:

·       Overall:

o   The manuscript needs significant editing for English language, grammar, typos, and consistency.

o   The manuscript needs to be better situated within the current literature, especially with regards to induction therapy, surgery, and de-escalation. The majority of current citations appear to be from when the study was initiated, over a decade ago. Phrasing such as “When this trial was initiated…” is honest and appreciated.

·       Methods:

o   Study design: this states that the patients were randomized according to smoking status. However, in Table 1, the smoking status appears to be quite uneven between Arms A and B. Could the authors comment and/or provider statistical testing of patient randomization? It also appears that the tumor distribution is uneven, with Arm B having more high-stage tumors.

Minor:

·       Abstract:

o   The abstract needs to be re-written to include the patient numbers, hazard ratios, and the death due to toxicity.

·       Introduction:

o   Define TPF

o   Add more citations to better contextualize the work within the current (as opposed to when the trial was initiated) landscape of HPV+ treatment

·       Figures:

o   Figure 1: include the specific regimens under Arm A and B in the actual figure

o   Figures 2 and 3: provide hazard ratios and confidence intervals at the pre-determined 2-year timepoints

·       Tables:

o   Table 1: consider adding statistical testing

o   Table 2: define acronyms in the legend

o   Table 3: include 8th edition staging as per Table 1

·       Methods:

o   Study design: is this the ACCROBAT I or II study? What does ACCROBAT stand for?

o   Patients:

§  How was resectability determined?

§  The trial appears to have been registered at the EU Clinical Trials Register and not Clinicaltrials.gov

o   Treatment: provide a justification or citation with regards to the induction chemotherapy protocol

o   Procedures during study/Follow-up: please provide a definition and reference to CTCAE v4.0

·       Results:

o   Provide insight as to recurrence patterns with regards to N stage.

o   Consider adding a waterfall plot of responses that are otherwise detailed as just text.

o   Include a table with toxicities.

o   Define the RTOG scale; provide references.

·       Discussion

o   Please discuss how many patients received the higher dose radiotherapy regimen in each arm. Did more patients receive a higher dosage in Arm B? How does this change interpretation of results?

o   How many patients’ tumors were PCR positive?

o   How many patients received carboplatin as opposed to cisplatin?

o   Please discuss the limitations of this study, including specific references to randomization, location, surgery, cetuximab (as opposed to cisplatin) in the chemoradiation phase, etc.

Round 2

Reviewer 3 Report

Mercke et al. have provided good rationale for their manuscript content choices in their response to this reviewer. They have made some edits to their manuscript; however, additional changes should be made prior to publication.

MAJOR:

-       Minimal changes were made to the English language in the reviews. Much of the manuscript still contains poorly written or confusing sentences, such as:

o   “The margin to the planning target volume (PTV) was generally 5 mm in all directions. Dose was prescribed so as 95% of that, was the best fit isodose to the PTV, with minor exceptions when dose to organs at risk (OAR) had higher priority.”

o   Such poor language needs to be addressed prior to publication.

-       Hazard ratios are needed for appropriate interpretation of the survival data. These were requested in the prior revision. The authors mentioned having added these in their reviewer response; but, they are nowhere to be seen. Hazard ratios should be present in Figures 2 and 3 in addition to the abstract. These should refer to the pre-determined 2-year timepoint. Progression-free survival percentages are not appropriate for a time series / Kaplan-Meier analysis.

MINOR:

-       The authors were recommended to add the specific regimens of arms A and B to Figure 1. These were requested in the prior revision. The authors mentioned having added these in their reviewer response; but, they are nowhere to be seen

-       Please define the ACCROBAT study in the text

-       Please include how resectability was determined in the text

-       Please clearly define TPF in the text. Although this is a well-established regimen (in the field), the authors do not define the acronym or specific regimen. A citation is not enough. This additional text is necessary for this broader-scoped journal.
